# Impact of Air Pollution and Allergic Status on Health-Related Quality of Life among University Students in Northern Thailand

**DOI:** 10.3390/ijerph21040452

**Published:** 2024-04-08

**Authors:** Tipanan Pisithkul, Tippapha Pisithkul, Mongkol Lao-Araya

**Affiliations:** 1Division of Allergy and Clinical Immunology, Department of Pediatrics, Faculty of Medicine, Chiang Mai University, Chiang Mai 50200, Thailand; riem.pisithkul@gmail.com; 2Program in Biotechnology, Faculty of Science, Maejo University, Chiang Mai 50290, Thailand; tpisithkul@gmail.com

**Keywords:** particulate matter, PM_2.5_, allergy, rhinitis, Thailand, well-being

## Abstract

Background: Global awareness of ambient air pollution has heightened due to its detrimental impact on health, particularly in regions with elevated PM_2.5_ levels. Chiang Mai has emerged as an area experiencing the highest PM_2.5_ levels in Thailand. Objectives: to examine the prevalence of respiratory allergies and assess the impact of air pollution on the health-related quality of life (QoL) among university students in Chiang Mai. Methods: Chiang Mai University (CMU) and Maejo University (MJU) students were recruited. The Global Asthma Network (GAN) questionnaire screened for respiratory allergies (RAs). The disease-specific QoL questionnaire (Rcq-36) was administered twice during low-PM_2.5_ and high-PM_2.5_ seasons to evaluate air pollution’s impact on health-related QoL. Those showing potential RAs underwent a skin prick test (SPT) to investigate allergic sensitization. Results: Out of 406 participants, 131 (32%) reported respiratory allergies. Among those undergoing SPT, a high rate (82.54%) had positive results. Across both universities, students reported significantly lower QoL in multiple domains, particularly respiratory, eye, sleep, and emotional well-being, during the high-PM_2.5_ season. This aligned with their poorer self-reported health on a visual analog scale (VAS; *p*-value < 0.01). PM_2.5_ levels significantly impacted social functioning for CMU students (*p*-value = 0.001) and role limitations for MJU students (*p*-value < 0.001). Notably, participants without respiratory allergies (non-RAs) were more significantly affected by PM_2.5_ than RA participants in almost all parameters, despite experiencing fewer baseline symptoms. Conclusions: Respiratory allergies, particularly allergic rhinitis, are prevalent among university students in Chiang Mai. This study underscores the substantial negative impact of ambient air pollution on QoL for both allergic and non-allergic students.

## 1. Introduction

Air pollution is a major public health concern, ranking high among environmental challenges. Within this landscape of environmental concerns, particulate matter (PM) emerges as a particularly potent and pervasive threat to human health. Airborne PM varies in size, composition, and is often classified by median aerodynamic diameter. Fine particulate matter less than 2.5 μm (PM_2.5_) is the most concerning and well-studied type of air pollution. These minute particles can infiltrate deep into the lungs, prompting grave concerns for a spectrum of health impacts across multiple organ systems. Oxidative stress plays a pivotal role in PM-mediated effects, contributing to systemic vascular dysfunction, cardiovascular, and respiratory complications [1,2,3,4].

In the context of global air pollution challenges, Thailand grapples with its distinctive environmental concerns, placing air pollution at the forefront of the nation’s most critical environmental issues. Air pollution is a chronic problem, particularly in the northern region of Thailand. Each year during the dry season (February–April), farmers burn fields to clear land, contributing to agricultural burning, with a higher risk of wildfires due to the vegetation type and climate. At the beginning of April, after the onset of rain, particulate matter dissipates, a pattern observed annually [5,6,7].

According to the Pollution Control Department (PCD) of Thailand, the highest daily average of PM_2.5_ concentrations in Northern Thailand was as high as 200–300 μg/m^3^ from February to April [8]. The severity of the situation is underscored by the fact that this concentration significantly exceeded the recommended guideline values, posing a considerable risk to public health. Regarding Thailand’s national ambient air quality standards (NAAQS), the 24 h average values should not exceed 37.5 μg/m^3^. The maximum daily average PM_2.5_ concentrations for Chiang Mai exceeded the standard from February to April 2023 for more than 70 days.

Respiratory allergies, encompassing allergic rhinitis (AR) and asthma, represent prevalent chronic non-communicable diseases affecting about one-third of the global population [9]. The prevalence of these diseases has increased in many parts of the world over the past decades, with an estimated global prevalence ranging from 10% to 30% [10,11,12]. The incidence varies by age, peaking in adolescents [12]. Notably, respiratory allergies are on the rise in Thailand. Previous studies, utilizing questionnaires and skin prick tests (SPT) for confirmation, have reported a prevalence as high as 58.5% among university students in Bangkok [13].

The key mechanism of allergic diseases involves cellular and tissue inflammation. The additional inflammation caused by ambient air pollution has the potential to damage tissues and exacerbate airway allergic symptoms [2]. As individuals worldwide navigate an environment saturated with pollutants, those with pre-existing respiratory allergies are particularly vulnerable to the exacerbating influence of PM_2.5_. Several studies have demonstrated the effects of PM_2.5_ on increased morbidity and mortality, emergency visits, and hospitalization for acute respiratory problems, including chronic obstructive pulmonary disease (COPD) and asthma, upper airway infections, and allergies [1,3,14]. Despite this, studies on the health-related quality of life (HR-QoL) in adolescents with respiratory allergies are limited. Therefore, the objectives of this study were to (1) examine the prevalence of respiratory allergies among university students in Chiang Mai; (2) investigate the impact of PM_2.5_ on the HR-QoL of students; and (3) compare the HR-QoL between allergic vs. non-allergic students, and compare different locations of the campus.

## 2. Materials and Methods

### 2.1. Ethics Statement

The study protocol was approved by the Research Ethics Committee, Faculty of Medicine, Chiang Mai University (protocol code 282/2565 and date of approval 15 August 2022). Informed consent was obtained from all participating students.

### 2.2. Study Population

The inclusion criteria were students aged 18–25 years from two universities in Chiang Mai, Northern Thailand: Health Science (Suandok) campus, Chiang Mai University (CMU), and Mae Jo University (MJU) in Sansai district, a suburban area. This study excluded participants with diagnosed chronic lung diseases (other than asthma), cardiovascular diseases, or a history of regular cigarette smoking.

The sample size calculation was based on a previous study reporting a 40% prevalence of allergic rhinitis [13], the sample size was estimated with a 5% margin of error before recruitment. This resulted in a minimum recruitment target of 368 participants. To account for an anticipated 40% non-response rate, 500 questionnaires were distributed. Participants were selected at random for their convenience.

Chiang Mai’s unique blend of urban and suburban areas creates dynamic variations in PM_2.5_ exposure for students. Dense urban centers, with their constant traffic flow and industrial activity, expose students to higher levels of fine particulate matter. This PM_2.5_ comes from sources like diesel vehicles and construction dust. In contrast, suburban areas, often a mix of residential zones and agricultural fields, present a different set of air pollutants. Rice paddies and agricultural burning practices contribute organic aerosols and ozone to the air [5,6,15].

Beyond these PM_2.5_ variations, lifestyle differences between urban and suburban students further influence their health experiences. Urban students may rely more on public transportation, leading to less outdoor time compared to their suburban counterparts. Additionally, suburban students, especially those attending MJU, a leading institution in agricultural studies, might engage in agricultural activities that expose them to different pollutants. 

### 2.3. Questionnaires

In this study, we employed the standardized written core questionnaires developed by the Global Asthma Network (GAN) [10] in 2016. These instruments had undergone rigorous translation, standardized format, and cultural adaptation for the Thai context [16]. Their comprehensive coverage of allergy-related information proved highly valuable. They not only explored asthma, rhinitis, and eczema symptoms in detail but also delved into aspects like doctor diagnoses, management strategies, and medication use.

The Rhinoconjunctivitis QoL questionnaire (Rcq-36) [17] is a comprehensive tool for assessing the HR-QoL in individuals affected by rhinoconjunctivitis. It consists of 36 questions that delve into six distinct categories: symptoms (including rhinitis, eye symptoms, and others), physical functioning, role limitations, sleep disruption, social functioning, and emotions. Additionally, two independent questions address overall health and work absences due to the disease. The explanation of the Rcq-36 domains is shown in the table in Appendix A. Each question was scored on a 0-to-4 scale, providing quantitative data on respondents’ experiences. This structure, combined with the questionnaire’s standardized translation and Thai validation, makes the Rcq-36 a reliable and nuanced tool for understanding the impact of rhinoconjunctivitis on individuals’ well-being [17].

The first wave of the questionnaire study was conducted during the low PM_2.5_ season (October to December 2022), followed by a second wave in the high PM_2.5_ season (February to April 2023). Students were invited to participate in both waves, firstly answering questionnaires about their baseline characteristics (health status, environment, activities, and demographics), and completing the GAN and Rcq-36 questionnaires for respiratory allergy screening and HR-QoL evaluation. During the second wave, the same participants were invited to respond to the Rcq-36 questionnaire again, alongside interviews that gathered additional information on their personal measurement and protection strategies during air pollution periods. Additionally, participants self-reported their overall well-being in both surveys using visual analog scales (VAS) [18]. This approach allowed us to capture changes in HR-QoL and personal protective behaviors in response to variations in PM_2.5_ levels.

### 2.4. Skin Prick Test

Participants reporting symptoms suggestive of respiratory allergies on the GAN questionnaires were further invited for investigation on allergen sensitization using skin prick tests (SPTs). The SPTs utilized commercially available extracts of common aeroallergens, including dust mites (*Dermatophagoides pteronyssinus* and *Dermatophagoides farinae*), mixed cockroach, para grass, careless weed, cat epithelium, dog epithelium, and *Cladosporium* spp. (AllerVACtest^®^, Bangkok, Thailand), using a lancet on the forearm. A positive test was considered if the weal diameter equaled or exceeded 3 mm compared to controls.

### 2.5. Air Quality Monitoring and Definitions 

Air quality monitoring data for PM_2.5_ were accessed through the Climate Change Data Center of Chiang Mai University website (https://www.cmuccdc.org/ accessed on 1 July 2023). The chosen monitoring stations were within 5 km of the universities and were deemed representative of the student residents’ exposure. The ‘low PM_2.5_ season’ was defined as an average 24 h PM_2.5_ level below 25 µg/m^3^ for at least 14 consecutive days. The ‘high PM_2.5_ season’ followed national standards, encompassing periods with either hourly PM_2.5_ levels exceeding 100 µg/m^3^ or 24 h averages above 50 µg/m^3^ for three consecutive days. 

AR was defined as experiencing nasal symptoms with or without eye symptoms, but excluding episodes attributed to “a cold or flu” in the past year among participants. This definition was based on specific GAN questionnaire wording addressing these symptoms [16]. Respiratory allergies encompassed participants who had either a doctor-diagnosed history of AR and/or asthma or who reported experiencing characteristic symptoms suggestive of these conditions.

### 2.6. Statistical Analysis

The analyses were performed using STATA/SE software (Stata/SE 14 for Windows, StataCorp LP, College Station, TX, USA). Results were expressed as mean ± SD, percentages, or 95% CI of responses to each question. We checked for data normality and determined parametric tests were appropriate for most comparisons. For non-normal data, Wilcoxon signed-rank tests were employed. Prevalence comparisons across seasons or groups were analyzed using chi-square tests. HR-QoL score changes within individuals across seasons were investigated using paired *t*-tests, while unpaired *t*-tests were used for comparisons between allergic and non-allergic groups, or across universities. A *p*-value < 0.05 was considered statistically significant.

## 3. Results

A total of 406 out of 500 completed questionnaires were returned (participation rate, 81.2%). The participants were almost equally distributed between CMU (*n* = 201) and MJU (*n* = 205). The baseline characteristics of the students who completed questionnaires in the repeated questionnaire surveys are shown in Table 1. Females constituted the majority (62%) and there was a significantly higher proportion of females in MJU compared to CMU. The mean ± SD age of participants was 19.7 ± 1.0 years old. Most of the CMU and MJU participants’ residential areas were in the inner-city and suburban areas, respectively, which were close to their universities’ locations. From the GAN questionnaire surveys, a total of 263 (64.8%) had reported rhinitis symptoms within the 12 months, and CMU students reported a significantly higher prevalence of self-reported rhinitis symptoms compared to MJU students (77% vs. 53%, *p*-value < 0.001). However, only 21.7% (*n* = 88) of participants met the criteria for ARs based on the GAN questionnaire. The prevalence of self-reported and doctor-diagnosed ARs and asthma were comparable between both universities (Table 1). The overall prevalence of respiratory allergies, including ARs and asthma, was 32.3%.

Among the 131 participants who reported symptoms suggestive of respiratory allergies, 48% (*n* = 63) underwent an SPT. Detailed results of the SPTs are presented in Table 2. Notably, 82.5% of these participants with self-reported respiratory allergy symptoms had positive SPT results. This might suggest a high correlation between reported symptoms and confirmed allergies in this population. The most common allergens identified were dust mites (*D. pteronyssinus* and *D. farinae*) and cockroaches, with 77.8% and 57.1% of students testing positive, respectively.

The average PM_2.5_ levels during the repeated survey periods, as measured by the nearest air quality monitoring stations to the universities, are shown in Table 3. During the low PM_2.5_ season, PM_2.5_ levels in CMU and MJU were similar. However, in the high PM_2.5_ season, the average PM_2.5_ level was significantly higher in Chiang Mai’s inner-city area (CMU) compared to the suburban area (MJU) (131 vs. 102 μg/m^3^, respectively, *p*-value < 0.001). Notably, the highest 24 h PM_2.5_ level recorded during the study period was 173.92 μg/m^3^ in March 2023.

Across both universities, students reported significantly lower HR-QoL in most Rcq-36 domains (increase in scores) particularly in respiratory, eye, sleep, and emotion domains, during the high-PM_2.5_ season compared to the low season (Table 4). This aligned with their poorer self-reported well-being health on VAS. Interestingly, MJU students spent more time outdoors, while CMU students engaged in more air-conditioned activities and utilized various personal protective measures. This suggests different coping strategies adopted by students, potentially highlighting a need for interventions promoting protective behaviors that balance outdoor activity with PM_2.5_ mitigation.

Comparing HR-QoL (Table 5), non-allergic participants reported significantly lower well-being VAS scores but higher Rcq-36 scores (poorer health and HR-QoL) in all domains, while allergic participants only had significantly higher Rcq-36 scores in role limitation, sleep, and social functioning. This likely reflects a higher baseline Rcq-36 score in allergic participants, potentially indicating pre-existing poorer symptoms that masked further changes. Interestingly, both groups engaged in similar outdoor activities and air conditioner use despite air quality differences. Analyzing QoL changes confirmed that non-allergic participants experienced a significantly larger decline in HR-QoL, particularly in symptom domains, during high PM_2.5_ seasons (Table 6).

## 4. Discussion

This study found a concerning prevalence of allergic rhinitis (AR) and respiratory allergies among university students in Chiang Mai, Thailand, affecting 22% and 32% of participants, respectively. Notably, over 80% of participants with self-reported respiratory allergies had positive skin prick tests, confirming their sensitization. Students’ HR-QoL significantly declined during the high-PM_2.5_ season, particularly in domains like respiratory symptoms, sleep disturbance, and emotional well-being. Participants without respiratory allergies were more significantly affected by PM_2.5_ than those with allergies in nearly all HR-QoL parameters, despite experiencing fewer baseline symptoms. This suggests that even individuals without diagnosed allergies may be vulnerable to the negative health effects of air pollution.

The 22% and 32% prevalence of AR and respiratory allergies, respectively, among our university students aligns with previous reports in general populations from the U.S.A. and Europe (20–30%) [9,11,12]. However, a 2014 study of Thai medical students in Bangkok using standardized methods reported a much higher AR prevalence at 58.5% [13]. This significant difference raises questions about potential regional variations in AR prevalence or the difference in study methodology. Furthermore, our study found that students residing in the inner-city area of Chiang Mai reported significantly more rhinitis symptoms and had a higher, though not statistically significant, AR diagnosis rate compared to suburban students. This suggests environmental factors may play a role in AR development and symptoms [12,16,19], although further research is needed. Future studies directly comparing environmental exposures and their impact on allergic symptoms across different regions would be valuable in understanding these variations.

Our study identified dust mites and cockroaches as the most prevalent aeroallergens, aligning with reports of common allergens among the Thai population [13]. The significant difference in sensitization rates to these allergens across different locations (CMU vs. MJU) likely reflects variations in living environments across residential areas [19]. Furthermore, 83% of participants with self-reported respiratory allergies had positive skin prick tests, suggesting the GAN questionnaire effectively identifies individuals with allergies. However, SPT testing was limited to participants with positive self-reported allergies, preventing the calculation of the questionnaire’s sensitivity and specificity. This hinders a definitive assessment of its diagnostic accuracy. Future studies evaluating the GAN questionnaire’s performance in diverse populations with confirmed allergies could provide valuable insights into its potential as a screening tool.

This study focused on the effects of PM_2.5_, a health-damaging component of outdoor air pollution, on university students’ HR-QoL. We found that students’ HR-QoL significantly declined during the high PM_2.5_ season, particularly in domains like respiratory and eye symptoms, sleep disturbance, and overall well-being. Interestingly, students at MJU (suburban area) experienced significantly worse role limitation symptoms (difficulty concentrating, easily tired, fatigue, headache, daytime sleepiness) compared to CMU (inner-city) students. This difference might be due to higher outdoor activity and less use of personal protective measures like masks and air purifiers in the MJU students. These findings suggest that PM_2.5_ exposure can negatively impact student health and well-being, with potential variations based on individual behavior and environmental factors [14].

Previous studies have highlighted the negative impacts of air pollution on allergic patients, including increased symptom severity, outpatient visits, and hospitalization rates [1,20,21]. To our knowledge, this is the first epidemiological survey in Thailand that used standardized questionnaires (GAN and Rcq-36) with a high response rate to assess the impact of air quality on university students’ HR-QoL. Our study found that non-allergic participants reported poorer overall well-being and HR-QoL during high PM_2.5_ seasons compared to low PM_2.5_ seasons, while this effect was less evident in allergic participants. This likely reflects a higher baseline symptom burden in allergic participants due to pre-existing symptoms or medication use, which might have masked further changes caused by air pollution. This finding emphasizes that even individuals without known allergies are susceptible to air pollution’s detrimental effects, potentially highlighting a hidden vulnerability in this population. It reinforces the need for broader public health interventions and awareness campaigns targeting the entire population, not just those with diagnosed allergies.

Our study has some limitations. Focusing solely on outdoor air quality might not capture the full picture, as students spend significant time indoors. Additionally, the outdoor air quality measurements may include not only PM_2.5_ but also coarse particulate matter (PM10) or other components. The short-term design limits our ability to understand long-term health effects, lag day from exposure before developing adverse symptoms, and generalize results across seasons. Surveys are inherently susceptible to recall bias, which could influence participants’ responses. Lastly, while self-reported allergies and positive questionnaire screening suggest potential respiratory sensitivity, the lack of objective confirmation (e.g., lung function testing) and low participation in SPT among screened individuals requires us to interpret the results cautiously.

## 5. Conclusions

This study identified a high prevalence of rhinitis symptoms and significant negative impacts of air pollution on university students’ HR-QoL in Chiang Mai. Notably, non-rhinitis participants experienced worse QoL during the high PM_2.5_ season, potentially due to their lack of awareness or protective measures. These findings highlight the need for broader interventions beyond school activities, such as public awareness campaigns, air quality monitoring, and community-based solutions to address PM_2.5_ sources. Protecting individuals, especially vulnerable groups, and tackling pollution at its core are crucial steps towards safeguarding health and well-being in this region.

## Figures and Tables

**Table 1 ijerph-21-00452-t001:** Baseline characteristics.

N(%)	Total (*n* = 406)	CMU (*n* = 201)	MJU (*n* = 205)	*p*-Value
Gender: male	115 (38.2)	94 (46.8)	61 (30.0)	<0.001
Age (mean ± S.D., years)	19.74 ± 1.03	19.85± 0.90	19.63 ± 1.13	0.03
Residential area:				<0.001
Inner-city	185 (45.6)	179 (89.1)	6 (2.9)	
Suburban	198 (48.8)	6 (3.0)	192 (93.7)	
Self-reported:				
Asthma	27 (6.7)	12 (6.0)	15 (7.3)	0.59
Allergic rhinitis	88 (21.7)	49 (24.5)	39 (19.0)	0.19
Rhinitis symptoms	263 (64.8)	155 (77.1)	108 (52.7)	<0.001
Doctor diagnosed:				
Asthma	26 (6.4)	10 (5.0)	16 (7.8)	0.66
Allergic rhinitis	36 (8.9)	21 (10.5)	15 (7.3)	0.22
Respiratory allergies	131 (32.3)	71 (35.3)	60 (29.3)	0.19

**Table 2 ijerph-21-00452-t002:** Skin prick test results (N(%)).

	Total	CMU	MJU
Participants (% with a history of respiratory allergies)	63 (48.1)	31 (43.7)	32 (53.3)
Gender: Male	28 (44.4)	17 (54.8)	11 (34.4)
SPT positives	52 (82.5)	28 (90.3)	24 (75.0)
*D. pteronyssinus*	49 (77.8)	27 (87.1)	22 (68.6)
*D. farinae*	49 (77.8)	28 (90.3) *	21 (65.6) *
Cockroach	36 (57.1)	18 (58.1) ^#^	18 (56.3) ^#^
Cat	19 (30.2)	13 (41.9)	6 (18.8)
Dog	4 (6.4)	3 (9.7)	1 (3.1)
Para grass	3 (4.8)	2 (6.5)	1 (3.1)
Careless weed	0 (0.0)	0 (0.0)	0 (0.0)
*Cladosporium* spp.	2 (3.2)	2 (6.5)	0 (0.0)

* Statistical significance, *p*-value = 0.018; ^#^ *p*-value = 0.045.

**Table 3 ijerph-21-00452-t003:** Average 24 h PM_2.5_ level (μg/m^3^) at nearest air quality monitoring station to universities (mean (min–max)).

	CMU	MJU
Low PM_2.5_ season (October–December 2022)	10.35 (9.34–14.67)	11.34 (10.45–18.03)
High PM_2.5_ season (February–April 2023)	131.38 (72.65–173.92) *	102.34 (74.75–148.40) *

* Statistical significance.

**Table 4 ijerph-21-00452-t004:** HR-QoL scores, personal activities, and air pollution protection measures among CMU and MJU students during low- and high-PM_2.5_ seasons (mean ± SD).

	CMU (*n* = 201)			MJU (*n* = 205)		
	Low PM_2.5_	High PM_2.5_	*p*-Value	Low PM_2.5_	High PM_2.5_	*p*-Value
Rcq-36 domains:						
Rhinitis	1.00 ± 1.04	1.16 ± 1.09	<0.001	1.05 ± 1.07	1.23 ± 1.03	<0.001
Eye symptoms	0.83 ± 1.04	1.05 ± 1.08	<0.001	0.93 ± 1.06	1.14 ± 1.09	<0.001
Other symptoms	1.10 ± 1.10	1.09 ± 1.12	0.38	1.25 ± 1.16	1.37 ± 1.17	0.001
Role limitations	0.66 ± 0.98	0.68 ± 0.97	0.16	0.64 ± 0.92	0.79 ± 0.95	<0.001
Sleep	0.38 ± 0.71	0.55 ± 0.91	<0.001	0.90 ± 1.14	1.05 ± 1.12	0.001
Social functioning	0.49 ± 0.84	0.62 ± 0.98	0.001	0.87 ± 1.04	0.94 ± 1.04	0.06
Emotions	0.65 ± 0.94	0.74 ± 1.02	<0.001	0.82 ± 1.06	0.93 ± 1.00	0.001
Overall health	1.42 ± 0.68	1.59 ± 0.76	<0.001	1.41 ± 0.81	1.52 ± 0.81	0.03
Well-being (VAS)	7.32 ± 2.28	6.41 ± 2.54	<0.001	7.02 ± 2.57	6.08 ± 2.77	<0.001
Average daily activity						
Outdoor (hr)	2.91 ± 2.93	2.61 ± 2.05	0.071	3.83 ± 3.68	4.40 ± 2.87	0.025
AC use (hr)	10.35 ± 5.71	11.61 ± 5.88	0.003	5.43 ± 4.49	5.22 ± 3.45	0.20
Personal air pollution protection measures (n(%))						
PM_2.5_ protection masks	39 (19.4)	99 (49.3)	<0.001	51 (24.9)	46 (22.4)	0.56
PM_2.5_ protection skin care products	5 (2.5)	15 (7.5)	0.002	16 (7.8)	14 (6.8)	0.70
Air purifier in bedrooms	61 (30.4)	114 (56.7)	<0.001	9 (4.4)	8 (3.9)	0.80
Use of AC in bedrooms	120 (59.7)	134 (66.7)	0.002	64 (31.2)	65 (31.7)	0.90

HR-QoL: health-related quality of life; Rcq-36: The Rhinoconjunctivitis QoL questionnaire; VAS: visual analog scale; AC: air conditioner.

**Table 5 ijerph-21-00452-t005:** HR-QoL scores and personal activities among respiratory allergic and non-respiratory allergic students during low- and high-PM_2.5_ seasons (mean ± SD).

	Respiratory Allergies (*N* = 131)			Non-Respiratory Allergies (*N* = 275)		
	Low PM_2.5_	High PM_2.5_	*p*-Value	Low PM_2.5_	High PM_2.5_	*p*-Value
Rcq-36 domains:						
Rhinitis	1.40 ± 1.17	1.42 ± 1.13	0.30	0.86 ± 0.96	1.09 ± 1.02	<0.001
Eye symptoms	1.22 ± 1.18	1.29 ± 1.11	0.09	0.73 ± 0.95	1.01 ± 1.06	<0.001
Other symptoms	1.43 ± 1.18	1.50 ± 1.23	0.04	1.06 ± 1.10	1.10 ± 1.10	0.05
Role limitations	0.82 ± 1.04	0.92 ± 1.05	0.003	0.58 ± 0.91	0.65 ± 0.90	<0.001
Sleep	0.82 ± 1.07	1.05 ± 1.16	<0.001	0.57 ± 0.95	0.69 ± 0.97	<0.001
Social functioning	0.84 ± 1.06	1.00 ± 1.12	0.002	0.61 ± 0.93	0.67 ± 0.96	0.04
Emotions	1.02 ± 1.10	1.08 ± 1.09	0.06	0.61 ± 0.94	0.72 ± 0.95	<0.001
Overall health	1.66 ± 0.84	1.77 ± 0.83	0.07	1.31 ± 0.69	1.45 ± 0.74	0.002
Well-being (VAS)	5.89 ± 2.51	5.53 ± 2.58	0.05	7.75 ± 2.20	6.57 ± 2.64	<0.001
Average daily activity						
Outdoor (hr)	3.60 ± 2.95	3.57 ± 2.78	0.46	3.27 ± 3.52	3.49 ± 2.59	0.16
AC use (hr)	8.36 ± 5.62	8.36 ± 5.73	0.42	7.71 ± 5.74	8.19 ± 5.73	0.09

HR-QoL: health-related quality of life; Rcq-36: The Rhinoconjunctivitis QoL questionnaire; VAS: visual analog scale; AC: air conditioner.

**Table 6 ijerph-21-00452-t006:** Changes in HR-QoL Scores * between high- and low-PM_2.5_ seasons among students with differences in universities and respiratory allergic status.

	CMU	MJU	*p*-Value	Allergic	Non-Allergic	*p*-Value
Rcq-36 domains:						
Rhinitis	0.15	0.17	0.27	0.10	0.26	<0.001
Eye symptoms	0.22	0.22	0.47	0.18	0.27	0.04
Other symptoms	−0.01	0.12	<0.001	0.01	0.14	0.002
Role limitations	0.02	0.15	<0.001	0.08	0.10	0.35
Sleep	0.17	0.15	0.34	0.19	0.11	0.09
Social functioning	0.13	0.07	0.17	0.13	0.05	0.07
Emotions	0.09	0.11	0.35	0.13	0.05	0.04
Overall health	0.16	0.11	0.23	0.14	0.13	0.15

HR-QoL: health-related quality of life; Rcq-36: The Rhinoconjunctivitis QoL questionnaire; * Rcq-36 score in high PM_2.5_ season—score in low PM_2.5_ season.

## Data Availability

The datasets generated during the current study are available from the corresponding author upon reasonable request.

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
