# Peer review of "Impact of Air Pollution and Allergic Status on Health-Related Quality of Life among University Students in Northern Thailand"

_ijerph, 2024, doi:10.3390/ijerph21040452_

Round 1

Reviewer 1 Report

Comments and Suggestions for Authors

Thank you for the opportunity to review this manuscript. The manuscript “Impact of air pollution and allergic status on health-related quality of life among university students in northern Thailand” presents an important topic. The study is well conducted and presented. I have suggested few points to be addressed for further improvement; mentioned below:

1. The study focuses on PM2.5 exclusively. However, it is still difficult to understand why the authors investigated only the fine particles and not the others. The authors have not discussed the other dust particles, particularly coarse particles. It would be good to mention details of PM10 in the introduction section with values and standards relevant to Thailand. Also, for comparative analysis discussion about PM10 will be of high importance.

2. Abstract Line 25 and 27: I would suggest adding analysis values of the results stated. It is difficult to understand the impact of results without values.

3. I would suggest replacing “Air pollution” and “Quality of life” in key words with similar words. As these words have already been used in the title of the manuscript.

4. It is always good to avoid the use of adjectives in scientific writings i.e., pervasive etc. etc.

5. Line 161: A total of 406 questionnaires were returned 201 (CMU) + 206 (MJU); MJU students will be 205 instead of 206?

Comments on the Quality of English Language

Addressed in comments for authors.

Author Response

  1. The study focuses on PM2.5 exclusively. However, it is still difficult to understand why the authors investigated only the fine particles and not the others. The authors have not discussed the other dust particles, particularly coarse particles. It would be good to mention details of PM10 in the introduction section with values and standards relevant to Thailand. Also, for comparative analysis discussion about PM10 will be of high importance.

Response: We have rewritten the introduction emphasis the important, standard national values and rationale of investigation of PM2.5. There are no standard regulation values of PM10 and other PM in Thailand. The discussion on the other than PM2.5 is also added in limitation (last paragraph of discussion).  

  1. Abstract Line 25 and 27: I would suggest adding analysis values of the results stated. It is difficult to understand the impact of results without values.

Response: The result part of abstract has been revised and the p-value had been added.

  1. I would suggest replacing “Air pollution” and “Quality of life” in key words with similar words. As these words have already been used in the title of the manuscript.

Response: “air pollution” and “quality of life” are replaced with “particulate matter” and “well-being”

  1. It is always good to avoid the use of adjectives in scientific writings i.e., pervasive etc. etc.

Response: Thank you for advice. The words have been replaced.

  1. Line 161: A total of 406 questionnaires were returned 201 (CMU) + 206 (MJU); MJU students will be 205 instead of 206?

Response: “206” has been replaced with “205”.

Reviewer 2 Report

Comments and Suggestions for Authors

This study examined the prevalence of respiratory allergies and assess the impact of air pollution on the health-related quality of life (QoL) among university students in Chiang MaiThailand. The comments are as follows:

1. The inclusion/exclusion standards of participants are not clear, especially the inclusion standard. Besides, the sampling process and calculation of sample size have not been explained.

2. Have the authors considered lag effect of air pollution on health-related quality QoL?

3. The manuscript needs extensive revision on its format, e.g., using three-line table, P-value should be in capital letter with italic format. The errors in the reference list should also be revised.

Comments on the Quality of English Language

Should be improved.

Author Response

This study examined the prevalence of respiratory allergies and assess the impact of air pollution on the health-related quality of life (QoL) among university students in Chiang Mai, Thailand. The comments are as follows:

  1. The inclusion/exclusion standards of participants are not clear, especially the inclusion standard. Besides, the sampling process and calculation of sample size have not been explained.

Response: The inclusion, exclusion criteria, sample size calculation, and sampling process have been additional described in “Materials and Methods”.  

  1. Have the authors considered lag effect of air pollution on health-related quality QoL?

Response: Due to limitation of the study design, we cannot evaluate the lag effect of air pollution on health-related quality QoL. Hence, we added this limitation in limitation part at the last paragraph of discussion.

  1. The manuscript needs extensive revision on its format, e.g., using three-line table, P-value should be in capital letter with italic format. The errors in the reference list should also be revised.

Response: Thank you for your advice. The error has been revised.

Reviewer 3 Report

Comments and Suggestions for Authors

The authors present the results of a health questionnaire and skin prick test on students in two universities in Thailand. The document is generally well-written and the authors were careful in their methodology and data analysis. Specific comments below:

The Introduction section would benefit from being divided into thematic subsections.

Line 13 (and elsewhere): Make sure to subscript the "2.5" in PM2.5.

Line 24: What is "emotions" in the context of this study?

Line 45: There is an extra space between "with" and "a".

Line 49 (and elsewhere): Make sure to superscript "3" in "m3".

Line 53: "Outrageously" is an opinion and should be excluded.

Line 67: "Milieu" is not a very common English word - consider "environment".

Line 97: It's unclear why suburban (and not rural) populations would engage in agriculture.

Line 124: It would be useful for the authors to include a reference to VAS.

The tables have bullet points at nearly every row, and these should be removed.

The Discussion section could benefit from being divided into subsections. For example, Limitations, Implications, etc.

It may be useful for the authors to include the questionnaire as an Appendix or Supplementary Information.

It would be good to discuss what other pollutants may be disparately elevated during both study periods. As was mentioned in the introduction, ozone affects suburban populations more and it should be explained as to whether this may affecting survey results.

Comments on the Quality of English Language

The English language for this article is of relatively high quality with only a few minor areas that could be improved.

Author Response

The authors present the results of a health questionnaire and skin prick test on students in two universities in Thailand. The document is generally well-written and the authors were careful in their methodology and data analysis. Specific comments below:

The Introduction section would benefit from being divided into thematic subsections.

Response: Thank you for your advice. The manuscript has been revised.

Line 13 (and elsewhere): Make sure to subscript the "2.5" in PM2.5.

Response: The error has been revised.

Line 24: What is "emotions" in the context of this study?

Response: The additional table for describing each domain of Rcq-36 has been added in appendix.

Line 45: There is an extra space between "with" and "a".

Response: The error has been revised.

Line 49 (and elsewhere): Make sure to superscript "3" in "m3".

Response: The error has been revised.

Line 53: "Outrageously" is an opinion and should be excluded.

Response: The error has been revised.

Line 67: "Milieu" is not a very common English word - consider "environment".

Response: The error has been revised.

Line 97: It's unclear why suburban (and not rural) populations would engage in agriculture.

Response: The additional explanations of relation between the areas and population have been added in study population part.

Line 124: It would be useful for the authors to include a reference to VAS.

Response: Reference of VAS has been added.

The tables have bullet points at nearly every row, and these should be removed.

Response: The error has been revised.

The Discussion section could benefit from being divided into subsections. For example, Limitations, Implications, etc.

Response: The study’s limitation has been divided in another paragraph in discussion. But, clinical implications and suggestion are in the conclusion part.   

It may be useful for the authors to include the questionnaire as an Appendix or Supplementary Information.

Response: Due to copy right issue, the questionnaires cannot be added. The original questionnaire can be found elsewhere as shown in the references. However, we added explanation of Rcq-36 domain in appendix table. 

It would be good to discuss what other pollutants may be disparately elevated during both study periods. As was mentioned in the introduction, ozone affects suburban populations more and it should be explained as to whether this may affecting survey results.

Response: These comments have been added in limitation.

Round 2

Reviewer 2 Report

Comments and Suggestions for Authors

The tables need to be drawn as three-line format. The manuscript needs proofreading before being accepted for publication.

Comments on the Quality of English Language

Need minor revision.